# When Familial Hearing Loss Means Genetic Heterogeneity: A Model Case Report

**DOI:** 10.3390/diagnostics11091636

**Published:** 2021-09-07

**Authors:** Camille Cenni, Luke Mansard, Catherine Blanchet, David Baux, Christel Vaché, Corinne Baudoin, Mélodie Moclyn, Valérie Faugère, Michel Mondain, Eric Jeziorski, Anne-Françoise Roux, Marjolaine Willems

**Affiliations:** 1Département de Génétique Médicale, Maladies Rares et Médecine Personnalisée, CHU Montpellier, Université de Montpellier, 34090 Montpellier, France; camille.cenni@chu-montpellier.fr; 2Laboratoire de Génétique Moléculaire, CHU Montpellier, Université de Montpellier, 34090 Monpellier, France; l-mansard@chu-montpellier.fr (L.M.); david.baux@inserm.fr (D.B.); christel.vache@inserm.fr (C.V.); corinne.baudoin@inserm.fr (C.B.); melodie.moclyn@inserm.fr (M.M.); valerie.faugere@inserm.fr (V.F.); anne-francoise.roux@inserm.fr (A.-F.R.); 3Service ORL, CHU Montpellier, Université de Montpellier, 34090 Montpellier, France; c-blanchet@chu-montpellier.fr (C.B.); m.mondain@chu-montpellier.fr (M.M.); 4Centre National de Référence Maladies Rares “Affections Sensorielles Génétiques”, CHU Montpellier, Université de Montpellier, 34090 Montpellier, France; 5INM, Université de Montpellier, INSERM U1298, 34090 Montpellier, France; 6Service de Pédiatrie Générale, Infectiologie et Immunologie Clinique, CHU Montpellier, Université de Montpellier, 34090 Montpellier, France; e-jeziorski@chu-montpellier.fr

**Keywords:** familial hearing loss, multiple diagnoses, non-syndromic hearing loss, *ACTG1*, *MYH9*

## Abstract

We describe a family with both hearing loss (HL) and thrombocytopenia, caused by pathogenic variants in three genes. The proband was a child with neonatal thrombocytopenia, childhood-onset HL, hyper-laxity and severe myopia. The child’s mother (and some of her relatives) presented with moderate thrombocytopenia and adulthood-onset HL. The child’s father (and some of his relatives) presented with adult-onset HL. An HL panel analysis, completed by whole exome sequencing, was performed in this complex family. We identified three pathogenic variants in three different genes: *MYH9*, *MYO7A* and *ACTG1*. The thrombocytopenia in the child and her mother is explained by the *MYH9* variant. The post-lingual HL in the paternal branch is explained by the *MYO7A* variant, absent in the proband, while the congenital HL of the child is explained by a de novo *ACTG1* variant. This family, in which HL segregates, illustrates that multiple genetic conditions coexist in individuals and make patient care more complex than expected.

## 1. Introduction

Hereditary hearing loss (HL) is the most common sensory-neural deficit and is characterized by a high degree of genetic and phenotypic heterogeneity. More than 70% of genetic HL is non-syndromic (non-syndromic hearing loss, NSHL) and can follow a pattern of autosomal recessive (AR) inheritance in 75–80% of cases, autosomal dominant (AD) inheritance in 20–25% of cases and X-linked inheritance in 1–1.5% of cases [1]. More than 120 responsible genes have been identified to date (https://hereditaryhearingloss.org/, last reviewed in 17 December 2020), of which there are at least 45 genes in patients with ADNSHL. Some of them are also involved in syndromic entities.

The development of massively parallel sequencing (MPS), which allows the study of thousands of genes simultaneously, whole exome and whole genome, has allowed one to highlight the concept of “atypical phenotype” as instances of dual or more molecular diagnoses [2]. The occurrence of multiple molecular diagnoses in a single individual has been reported in 2–7.2% of cases [2,3].

In this article, we describe a family with both HL and thrombocytopenia caused by pathogenic variants in *MYO7A*, *ACTG1* and *MYH9* genes.

## 2. Materials and Methods

### 2.1. Clinical Report

The proband was 3 years old when referred, issued from unrelated Caucasian parents (Figure 1A). She was born at term after a normal pregnancy, without cytomegalovirus infection. All birth parameters were normal (50th centile), and otoacoustic emissions were present at birth. She presented neonatal thrombopenia (between 1000 and 3000 platelets/mm^3^) with subependymal and retinal hemorrhages without clinical severity, treated by two platelet transfusions and one immunoglobulin infusion. At 6 months, she had a normal platelet count (160,000/mm^3^). She was able to walk at 18 months and had normal development. An absence of language at 18 months led to the diagnosis of evolutive moderate-to-severe sensorineural HL (Figure 1B). A subsequent analysis at 2 and 2.5 years confirmed an asymptomatic and chronic thrombocytopenia. At 7 years and 9 months of age, she weighed 24.5 kg (70th centile), measured 126 cm (75th centile) and her head circumference was 52.5 cm (30th centile). She presented some ecchymoses but no severe hemorrhages, a ligamental hyper-laxity and advanced myopia (−7 diopters in the right eye and −10.5 diopters in the left eye) associated with large optic discs.

In the paternal branch, at least five men over two generations presented with evolutive, sensorineural HL, with an onset in the third decade (Figure 1A,B), suggesting an AD sensorineural HL inheritance. Four women, from three generations, in the maternal branch, presented with non-severe thrombopenia, between 20,000 and 100,000 platelets/mm^3^, associated with a sensorineural HL, with an onset of around 40 years of age (Figure 1A,B). The proband’s mother developed mild deafness at 40 years old.

### 2.2. Molecular Analysis

Informed consent for genetic analysis was obtained from the family in compliance with national ethics regulations. DNA from all affected members was isolated from peripheral blood samples by standard procedures.

#### 2.2.1. NSHL Gene Panel Sequencing

Both proband and father underwent MPS gene panel testing. In total, 74 NSHL genes were screened using the NimbleGen SeqCap EZ Choice technology [4]. Each exon (coding and non-coding) and its surrounding 50 bp intronic sequences, referenced in RefSeq or Ensembl, was targeted. Sequencing was performed on an Illumina MiSeq instrument (version 2 chemistry), and we used the MiSeqReporter software (v2.5) for the secondary analysis of the generated data. Variant calling files (VCFs) were automatically included in our in-house database system (USHVaM2), which also handled variant annotation. Lastly, variants of interest were confirmed by Sanger sequencing, and segregation analysis was performed on available members of the family.

#### 2.2.2. Whole Exome Sequencing (WES)

Genomic DNA was obtained from blood samples belonging to the proband and her parents. Library preparation was performed with the Nimbelgen SeqCap EZ MedExome kit (Roche Technology) according to the manufacturer’s instructions. Exome-enriched libraries were sequenced using the Illumina NextSeq system (Illumina, San Diego, CA, USA). Bioinformatic analysis of sequencing data was based on an in-house pipeline (https://github.com/beboche/nenufaar, accessed on 17 December 2019) generate a merged BAM and VCF file for the family. Quality data revealed more than 91% of the target nucleotides covered at 30X for individuals with an average coverage of 120X. Tertiary analysis involved the MobiDL captainAchab workflow (https://github.com/mobidic/MobiDL, accessed on 17 December 2019), based on ANNOVAR [5], MPA [6] and Captain-ACHAB (https://github.com/mobidic/Captain-ACHAB, accessed on 17 December 2019).

## 3. Results

The gene panel study identified two heterozygous variants in *MYH9* and in *ACTG1* in the proband (Figure 1A).

The c.3493C>T; p.(Arg1165Cys) variation in *MYH9* (NM_002473.5) has already been reported to be implicated in *MYH9* syndromic disease [7]. The mother and maternal grandmother were also heterozygous for this variant. The c.721G>A; p.(Glu241Lys) variant in the *ACTG1* gene (NM_001614.4) has already previously been detected in a family with ADNSHL [8]. This variant was absent in the parents, supporting a de novo occurrence, although a germline mosaicism could not be excluded. Gene panel testing was then performed for the father to elucidate the origin of his HL, and the c.2767_2769del; p.(Lys923del) variation in the *MYO7A* gene (NM_000260.3) was identified. It was absent from the control population databases (GnomAD, dbSNP). Sanger analysis confirmed the segregation of the variant in all the hearing-impaired members of the paternal branch. It is classified as likely pathogenic according to ACMG criteria [9]. We submitted this variant to the ClinVar database in NCBI (www.ncbi.nlm.nih.gov/clinvar/, accession number: VCV000930183.1, accessed on 19 June 2020).

WES was performed on the trio to explain the hyper-laxity and the myopia diagnosed in the proband, but no additional variants of interest could be identified.

## 4. Discussion

In conclusion, alterations in three different genes are responsible for the symptoms present in this family. The thrombocytopenia and the HL segregating in the maternal branch are explained by the *MYH9* variant. The child’s HL is explained by the de novo *ACTG1* variant, and the HL on her father’s side is explained by the *MYO7A* variant.

We report a new variant in the *MYO7A* gene, p.(Lys923del), resulting in an in-frame loss of a conserved lysine residue at codon 923, which segregates with the disease in this family, compatible with AD transmission. The *MYO7A* (*276903) encodes an unconventional myosin and is expressed in the pigment epithelium, the photoreceptor cells of the retina and the human embryonic cochlear and vestibular neuroepithelia [10]. Pathogenic alterations have been reported to cause syndromic HL (Usher syndrome type 1B, #276900) [10] or NSHL (DFNB2, #600060 and DFNA11, #601317) [10,11]. DFNA11 is characterized by a symmetric and progressive neurosensory HL with post-lingual onset. However, the degree of HL can be significantly different within the same family or among patients in the same age group [11]. Liu et al. had previously reported a family with DFNA11 and an in-frame 9-bp deletion leading to the loss of three residues, including two lysines at codons 887 and 888 [11]. Both this mutation and our variant are located in the same single alpha-helix (SAH) region. The SAH regions are rich in charged residues, which are predicted to stabilize the alpha-helical structure by ionic bonds and are constant force springs in proteins [12]. We concluded that p.(Leu923del) is a new dominant pathogenic variant.

The *ACTG1* gene (*102560) encodes actin gamma 1, a member of a highly conserved cytoskeletal protein family that plays fundamental roles in nearly all aspects of eukaryotic cell biology. This protein is particularly abundant in the specialized hair cells of the inner ear. *ACTG1* was identified as a causative gene for ADNSHL (DFNA20/26, #604717) [13]. Patients present with progressive post-lingual HL, and the age of onset ranges between the first and the fourth decades. Very early-onset or pre-lingual-onset HL has already been described [13]. *ACTG1* is also implicated in Baraitser–Winter syndrome type 2 (BWS2, #614583), which is a rare developmental syndrome characterized by dysmorphism traits, intellectual disability and congenital anomalies [14]. However, no genotype–phenotype correlation has been made to explain both syndromic and non-syndromic diseases linked to *ACTG1* [13]. We reviewed the *ACTG1* variants to compare the position in the gene and in the protein domains, the difference between the wild-type and the mutated residues, the impact on the conformation and the protein interactions. Any of the variants implicated in one or the other diseases are heterozygous missense. We compared 35 different variants: 11 implicated in the BWS2 (31%) and 24 in the DFNA20/26 (69%). Variants are distributed over the entire gene without regional clusters (Figure 2). No variant is involved in both BWS2 and DFNA20/26. Interestingly, two different variants alter the same residue: p.(Glu334Gln) involved in BWS2 and p.(Glu334Asp) in DFNA20/26. We evaluated the possible alteration of the conformation and the interactions with other molecules through the Hope3D website (https://www3.cmbi.umcn.nl/hope/input/, accessed on 19 June 2020) [15]. We found that more 8/11 BWS2 variants (73%) than 9/24 DFNA20/26 variants (37.5%) could alter the conformation of the gamma-actin based on the residues’ size or polarity difference. We also found that only 1/11 BWS2 variants (9%) whereas 9/24 DFNA20/26 variants (37.5%) could alter the “predicted” interaction between the protein and its ligands. However, the functional consequences of *ACTG1* heterozygous missense in BWS2 and DFNA20/26 still remain unclear. Rivière et al. suggested that BWS2 represents the severe end of a spectrum of cytoplasmic actin-associated phenotypes that begins with DFNA20/26 and extends to BWS2 [16].

We found no explanation for the severe myopia and hyper-laxity. BWS2 is associated with ocular features, including microphthalmia or coloboma [17], and articular features with progressive joint stiffness, which are not similar to those observed in our patient. Therefore, we cannot conclude whether the hyper-laxity and myopia observed in our patient are related to the extra-auditory features from the *ACTG1*-related disorder.

## 5. Conclusions

Prior to molecular investigations, we postulated that the proband had inherited two AD adulthood-onset HL defects, explaining a more severe and childhood-onset phenotype. However, we were surprised to find a de novo *ACTG1* variant as the cause of her HL.

In this study, we show that it is necessary to perform systematic and unbiased molecular studies in several individuals within a family when phenotypes resemble one another but are not the same. This is especially true in pathologies such as HL, which is frequent and genetically heterogeneous.

## Figures and Tables

**Figure 1 diagnostics-11-01636-f001:**
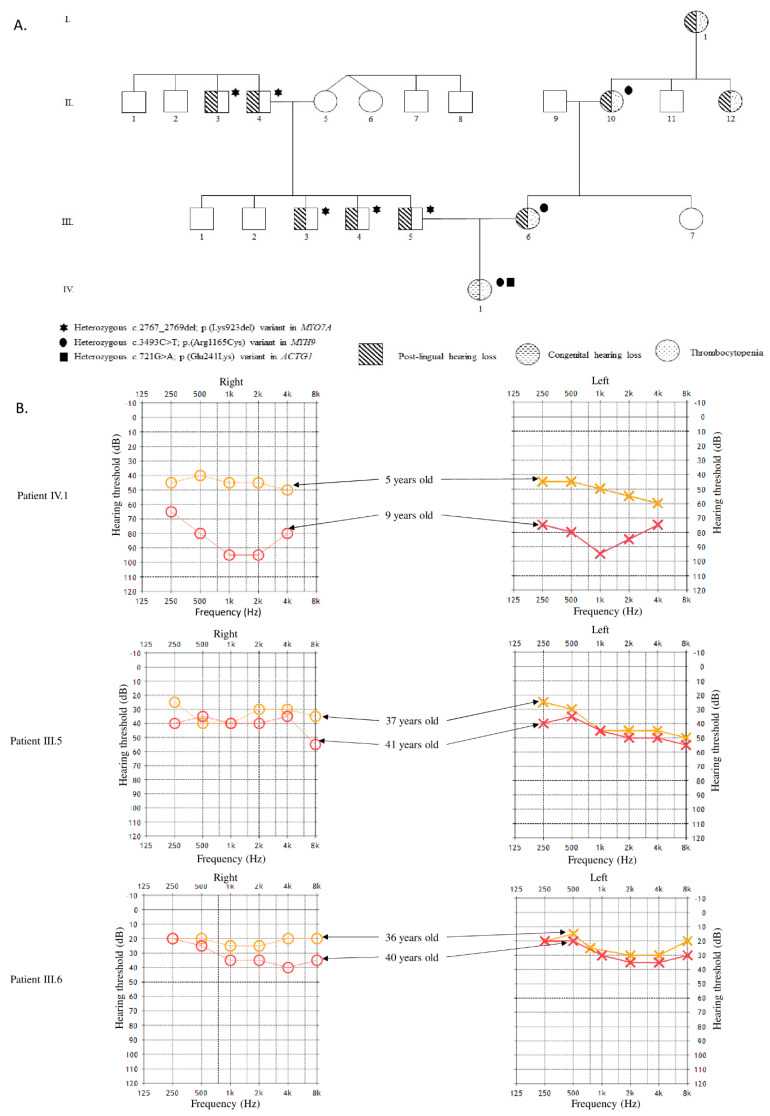
(**A**): Pedigree of the family. (**B**): Audiograms of proband and her parents.

**Figure 2 diagnostics-11-01636-f002:**
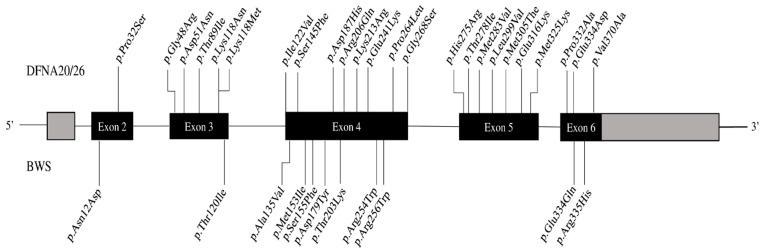
*ACTG1* variants. Schematic representation of the full-length *ACTG1* gene showing the different exons and locations of the reported variants. Variants associated with DFNA20/26 are indicated above the gene and those associated with BWS2 are below the gene.

## Data Availability

Not applicable.

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
