# Peer review of "When Familial Hearing Loss Means Genetic Heterogeneity: A Model Case Report"

_diagnostics, 2021, doi:10.3390/diagnostics11091636_

Round 1

Reviewer 1 Report

Camille Cenni et al present a family with both hearing loss and thrombocytopenia, caused by pathogenic variants in three genes as an example of “atypical phenotype” with dual or more molecular diagnoses. They demonstrate necessity to perform systematic and unbiased molecular studies in several individuals within a family when phenotypes resemble one another but are not the same, especially in frequent and genetically heterogeneous diseases.

The methods of the study are adequately described, and the conclusions are supported by the clearly presented results.

I suggest that the article may be published as it is, only with a minor spell check.

Reviewer 2 Report

Cenni et al. reported the importance of a whole-exome analysis in addition to a classical panel analysis in the complex familial hearing loss. They found a de novo mutation in ACTG1 gene in the proband and concluded that the de novo mutation is responsible for congenital hearing loss of the proband.

The work is interesting, and the manuscript is well written. The reviewers agree with the authors’ conclusion.

Reviewer 3 Report

The manuscript entitled “When familial hearing loss means genetic heterogeneity: a model case report” has discussed a rare hereditary hearing loss case from a complex family. They described three pathogenic variants in three different genes: MYH9, MYO7A and ACTG1 through whole-exome sequencing analysis. The data was well collected and presented. It illustrated the complexity of HL in separate patients and families. This work showed us that it was necessary to perform systematic and unbiased molecular studies in several individuals within a family even when phenotypes seemed similar, especially in those heterogeneous diseases. However, it’s better to adjust the resolution of the figures to make the data much clearer.